# COHEN WELLING BASES & $SO(2)$-EQUIVARIANT CLASSIFIERS USING TENSOR NONLINEARITY.

## ABSTRACT

In this paper we propose autoencoder architectures for learning a Cohen-Welling (CW)-basis for images and their rotations. We use the learned CW-basis to build a rotation equivariant classifier to classify images. The autoencoder and classifier architectures use only tensor product nonlinearity. The model proposed by Cohen & Welling (2014) uses ideas from group representation theory, and extracts a basis exposing irreducible representations for images and their rotations. We give several architectures to learn CW-bases including a novel *coupling* AE architecture to learn a coupled CW-bases for images in different scales simultaneously. Our use of tensor product nonlinearity is inspired from recent work of Kondor (2018a). Our classifier has very good accuracy and we use fewer parameters. Even when the sample complexity to learn a good CW-basis is low we learn classifiers which perform impressively. We show that a coupled CW-bases in one scale can be deployed to classify images in a classifier trained and tested on images in a different scale with only a marginal dip in performance.

## 1 INTRODUCTION

A fundamental problem in vision is understanding how the human eye sees objects and images as being the same even when they undergo transformations. To obtain such a behaviour in a machine learning set-up, a natural idea is to construct representations of the object which remain the same even when the object undergoes transformations. This 'handcrafting' of representations of the object *invariant* to transformations was the preferred method of the vision community for a long time. Since the pathbreaking achievements of modern convolutional neural networks beginning with the seminal work of Krizhevsky et al. (2012) the focus has changed - the interest is more now on developing algorithms that learn to construct these invariant representations from transformed examples.

Goodfellow et al. (2009) were among the first to address the question of invariance in deep networks. Cohen & Welling (2014) were among the first to use representation theory of compact Lie groups to give it a sound mathematical framework. Developing on on ideas from earlier works of Rao & Ruderman (1999) and Sohl-Dickstein et al. (2010) on the Lie group model, Cohen and Welling build a model for the abelian group of transformations $SO(2)$, the group of rotations of the plane. Using their model they learn a nice basis for the underlying vector space in which the images sit, which allows them to read off the types and multiplicities of the irreducible representations of $SO(2)$ which occur in images under rotations. In this *basis exposing irreducible representations*[1], the action of an element of $SO(2)$ on an image is very easy to describe. From projections of images onto this basis,[2] Cohen and Welling obtain features of images which they use for classification.

A number of experts attribute the impressive performance of modern deep convolutional networks (CNN's) to the fact that more complex and abstract features are learned as one moves down the network. Nonlinearity seems to be essential to learn such features. Deep networks achieve their nonlinearity by the use of activation functions such as RELU, Krizhevsky et al. (2012). Convolution networks are by design invariant to translations. Given the incredible success of these deep networks the quest is now to build networks that have invariance to a larger group of symmetries.

---

[1] a phrase from Cohen & Welling (2014)

[2] hereafter called a CW-basis

In Bruna & Mallat (2013) the authors show that the first two layers of the scattering network yield a powerful representation of images which is invariant to geometric transformations. Gens & Domingos (2014) proposed deep symmetry networks, symnets, that form feature maps over large symmetry groups. They showed that Symnets over the affine group have smaller sample complexity. More recently, Group Convolutional Networks (GCNN's) were introduced in Cohen & Welling (2016a) and Harmonic nets were introduced by Worrall et al. (2017). Both GCNN's and Harmonic nets are designed to learn representations of images invariant to a larger set of symmetries than translations. The resulting networks perform impressively - the error rate reported by GCNN's on the MNIST-rot(MNIST Variations) data set is only 2.28%. Harmonic nets are now the state of the art on MNIST-rot with a reported 1.62% error.

More recently Cohen & Welling (2016b) introduced Steerable CNN's based on more sophisticated ideas from group representation theory. This puts constraints on the network weights and architecture, and results in a reduction in the number of parameters to be learned. Steerable CNN's were shown to outperform ResNets, He et al. (2016), and achieve state of the art results on CIFAR 10,100.

In both GCNN's as well in in steerable CNN's equivariance to the symmetries of a group $G$ is achieved by a generalized form of convolution. The standard convolution in CNN's is replaced with the convolution of functions from $G$ to $\mathbb{C}$. Kondor & Trivedi (2018b) proved the converse - they showed that any neural network which achieves equivariance with respect to the action of a group $G$, necessarily implements such a generalized convolution of functions on $G$.

If one were to factor out depth, it appears that it is the nonlinearity of the activation functions used in deep networks that is responsible for their impressive performance. But nonlinear functions like RELU are mathematically hard to analyze. It would be nice to design networks which construct features, just as deep networks seem to do in their various layers, with more mathematically amenable activation functions. This is the first motivation of this study.

Our other motivation comes from what is done in Pattern theory, Mumford (1997), where to learn images a stochastic model of the images to be learned is first built. To build a stochastic model of natural images it is important to view images under different scales simultaneously (Mumford (1997, Chapter 6)). Scattering networks address this issue because they use a wavelet basis for images.

This leads us to the following motivating questions. Can one use a mathematically amenable nonlinearity like tensor-product nonlinearity to design neural nets for image classification, leveraging ideas from group representation theory? How does one obtain complex abstract features from simple features using such a nonlinearity? Does it help to view images under different scales simultaneously to build such neural nets?

In this paper we make a modest attempt to answer such questions. We combine ideas from Cohen & Welling (2014) and Kondor (2018a), and design neural networks leveraging notions of tensor products of group representations. We decouple the classification process in two steps - first discover Cohen-Welling bases and then deploy them to train classifiers using ideas from Kondor.

## 2 PRELIMINARIES

We start with some definitions which set the stage for the work of Cohen & Welling (2014). Let $G$ be a group and let $V$ be a finite-dimensional vector space over a field - we will always assume that the underlying field is the field of complex numbers $\mathbb{C}$, or the field of real numbers $\mathbb{R}$. Let $GL(V)$ denote the group of invertible linear transformations of $V$.

**Definition 1** *We say $G$ acts on $V$ if there is a group homomorphism $\rho$ from $G$ to $GL(V)$.*

So $g \in G$ acts on a vector $v \in V$ by sending $v$ to $\rho(g)v$. If $V$ has dimension $d$, then with respect to a basis of $V$, $\rho(g)$ is given by a $d \times d$ matrix. The action is then given by multiplying $\rho(g)$ with $v$. We denote this by $g \cdot v$.

**Definition 2** *We say a subspace $W$ of $V$ is $G$-invariant under the action of $G$ if for all $g \in G$ and all $w \in W$, $\rho(g)w \in W$. If $V$ has no proper subspace which is $G$-invariant we say $V$ is irreducible. The restriction of the action of $G$ to a $G$-invariant subspace $W$ is called a subrepresentation.*

Suppose $V$ has an invariant subspace $W$, and we can also find a $G$-invariant subspace $U$ such that $W \oplus U = V$. Then one can choose a basis of $W$ and of $U$ giving us a basis of $V$. In this basis each $\rho(g)$ will be a block diagonal matrix consisting of two blocks of size $dim(W)$ and $dim(U)$. For certain groups (called reductive groups) there is always such a $U$, and the group of rotations of the plane, $SO(2)$, is an example of such a group. For reductive groups we can continue this process till we have a decomposition of $V$ into irreducibles, $V = U_1 \oplus U_2 \oplus \ldots \oplus U_k$.

**Definition 3** *If $V$ and $W$ are representations of a group $G$, a $G$-morphism (aka a $G$-equivariant map) from $V$ to $W$ is a linear map $\phi$ from $V$ to $W$ such that $\phi(g \cdot v) = g \cdot \phi(v)$.*

If $V$ and $W$ are vector spaces over complex numbers Schur's lemma, Serre (1977), places restrictions on the dimension of the space of $G$-morphisms. First, if $V$ and $W$ are irreducible representations of $G$, Schur's lemma states the space of $G$-morphisms between them is either one dimensional (i.e every $G$-morphism between them is given by a scalar matrix) or zero dimensional (i.e the zero morphism is the only $G$-morphism between them). This allows us to group irreducible representations into types, with two of them being of the same type when there is a nonzero $G$-morphism between them.

Given a decomposition of a vector space with a $G$-action into a direct sum of irreducibles as above, we collect all irreducibles of the same type together and write $V = m_1 S_{t1} \oplus m_2 S_{t2} \cdots \oplus c_k S_{tk}$. Here $tk$ is an indexing set for types, $S_{ti}$ is an irreducible $G$-representation of type $ti$, and $S_{ti}$ and $S_{tj}$ are of different types when $i \neq j$. We then say $S_{ti}$ occurs in $V$ with multiplicity $m_i$ in the decomposition of $V$. Now assuming that $V = \oplus_{i=1}^{i=k} m_i S_{ti}$ and $W = \oplus_{i=1}^{i=k} n_i S_{ti}$, it follows that the dimension of the space of $G$-morphisms between $V$ and $W$ is $\sum_i m_i n_i$.

Cohen & Welling (2014) considered the problem of supervised learning of images under the group $SO(2)$. Viewing an $N \times N$ image as an $N^2$ dimensional vector, an element of $SO(2)$ acts on an $N^2$ dimensional vector by taking the vector representing the image to the vector representing the rotated image. $SO(2)$ is an abelian group (rotating an image by $\theta_1$ and then by $\theta_2$ or rotating it first by $\theta_2$ and then $\theta_1$ yields the same image). Hence, over complex numbers, the irreducible representations of $SO(2)$ are one-dimensional (see, Kanatani (1990)). When the underlying space is a real vector space (as was considered in Cohen & Welling (2014)), the types are parametrized by nonnegative integers $n$ (over complex numbers the irreducible representations are parametrized by all integers). The irreducible representation corresponding to type $n = 0$, $S_0$ is one-dimensional and is an invariant under $SO(2)$. For $n > 0$, the irreducible representation $S_n$ is two dimensional (over complex numbers this splits into two one-dimensional representations parametrized by $\pm n$).

It follows, say, from Kanatani (1990, 2.3.7, 2.4.7) that there is a change of basis of the underlying vector space $V$ with respect to which the action of the rotation group is given by a block diagonal matrix $D$ with blocks of size 1 and 2. It is easy to describe these blocks. Blocks of size 1 correspond to invariants, of type $n = 0$, and have entry 1. The number of such blocks is called the multiplicity of the space of invariants in $V$. The block corresponding to nonzero $n$ is the elementary rotation matrix $\begin{bmatrix} \cos(n\theta) & -\sin(n\theta) \\ \sin(n\theta) & \cos(n\theta) \end{bmatrix}$. It follows that the number of such blocks is the multiplicity of irreducible type $n$ in $V$.

So there exists an $N \times N$ orthogonal matrix $W$ and a diagonal matrix $D$, as above, with the property that when an image $x$ is rotated by $\theta$ the transformed image is given by $WDW^t x$. In Cohen & Welling (2014), the authors address the question of determining the matrix $W$ and the multiplicity with which a given $n$ appears. The input is collection of pairs $(x, y)$, of an image $x$ and it's transformation $y$ when it is rotated by an unknown angle $\theta$. The authors find $W$ using expectation-maximization over an interesting prior. The $W$ their algorithm outputs is almost always a small subspace of the image space, spanning only some of the irreducible subspaces (similar to an SVD procedure which just outputs only the top few singular vectors). So strictly speaking we should be calling this a CW-subbasis since it may not span the image space, but we ignore this technicality.

An image $x$ is then projected onto the $W$ constructed and the vector of projected values, $W^T x$, were used by Cohen and Welling as elementary features of the image for classification.

Recently Kondor (2018a) proposed a neural network for learning the behaviour and properties of complex many-body physical systems. The neurons in this system operate entirely in the Fourier space. The neurons compose activations following Schur's lemma, thereby ensuring that the activations are covariant to rotations.

We pose our motivating questions more precisely using the language of Cohen-Welling.

1. Can we obtain a CW-basis of images and their rotations using tensor product nonlinearity?

2. How can elementary features obtained using a CW-basis be combined to get more complex abstract features for building good classifiers?

3. Can one discover CW-bases of images in two scales in tandem, with one influencing the discovery of the other? Do such *coupled* bases perform better in classification? Can one use these *coupled* CW-bases interchangeably?

*Our contributions* We give an affirmative answer to all the questions. We give several architectures to construct such bases. We use Kondor's framework to construct a classifier which takes elementary features and combines them using simple ideas from the representation theory of $SO(2)$.[3] We give a simple coupled autoencoder architecture which answers question 3 above - we use tensor product nonlinearity crucially. While such bases are not better at classification we show that they can be interchangeably used with a marginal drop in performance.

## 2.1 Notions from representation theory

To describe our set up and explain our experiments we will need a few more notions from representation theory. Throughout we will assume that the underlying group $G$ is a reductive group, in fact the reader may think of $G$ as $SO(2)$.

If $V$ is a representation of a group $G$ then the dual vector space $V^*$ acquires a natural $G$ action given by $(g \cdot f)(v) = f(g^{-1}v)$, for $f \in V^*, v \in V$. If $V, U$ are representations of a reductive group then $V \otimes U$ is a representation of the product group $G \times G$. Furthermore, since $G$ is a subgroup of $G \times G$ under the diagonal embedding ($g$ mapping to $(g, g)$), $V \otimes U$ is a representation of $G$. And by what we described above $V \otimes U$ splits into a direct sum of irreducible representations of $G$ identified by their types and multiplicities. It also follows that the tensor algebra, $T(V) = \oplus_i V^{\otimes i}$ of $V$, is an (infinite dimensional, algebraic) representation of $G$.

**Remark 4** *When $G$ is $SO(2)$ and $0 \neq i < j$ then $S_i \otimes S_j$ splits into a direct sum $S_{i+j} \oplus S_{j-i}$. When $i = 0$, $S_i \otimes S_j$ is isomorphic to $S_j$, (Kanatani (1990)).*

To make precise the question about coupled bases and features from a coupled bases we need one more definition.

**Definition 5** *We say $G$-representations $V$ and $W$ are coupled if $W$ is the image of a $G$-morphism of a finite dimensional subrepresentation of the tensor algebra of $V$, and $V$ is the image of a $G$-morphism of a finite dimensional subrepresentation of the tensor algebra of $W$.*

The above definition is equivalent to requiring that $W$ can be realized as a subrepresentation (or a quotient) of the tensor algebra of $V$ and $V$ can realized as a subrepresentation (or a quotient) of the tensor algebra of $W$.

**Definition 6** *Let $V$ be the vector space of $14 \times 14$ images and let $W$ be the vector space of $28 \times 28$ images. Assume the rotation group $SO(2)$ acts on both. We say a Cohen-Welling basis $X$ of $V$ is coupled to a Cohen-Welling basis $Y$ of $W$ if the vector space dual of the subspace spanned by $X$ (with its $SO(2)$-action) is coupled to the vector space dual of the subspace spanned by $Y$.*

Recall that by projecting an image onto a CW-basis we get elementary features of the image which are used for classification. These elementary features are elements of the vector space dual of the space spanned by the CW-basis, hence the need for the word dual in the above definition. Motivated by the above definition we say features obtained from coupled CW-bases are *coupled features*.

The definition above is motivated by our question of whether one can build CW-bases in two different scales with one influencing the discovery of the other and vice-versa. Our definition suggests

---

[3]As we were getting this write up ready we were pointed to some recent work of Kondor et al. (2018c) - the authors do similar work, but use the rotation group $SO(3)$ - for completeness we have included their results in Table 1.

that in order to generate coupled CW-bases we will need to view the images at different scales simultaneously, and use tensor product nonlinearity to generate them, thereby forcing the influence we are looking for.

**Remark 7** *In the appendix we describe our problem and solution in a language that is more familiar to the ML community - as that of seeking filters in the Fourier space of images.*

**Remark 8** *One last remark is in order before we describe our setup and experiments. We are working over the field of real numbers, so Schur's lemma does not apply in our situation. Nevertheless when we look for $SO(2)$-morphisms between two irreducible vector spaces with an $SO(2)$-action we will assume that the morphisms are of the kind Schur's lemma dictates - either we have no morphism or the morphism is given by a scalar matrix $r Id$, $r$ a real number and $Id$ being the identity matrix.*

## 3 EXPERIMENTS AND RESULTS

We use the notation $W_\ell$ for a CW-basis of images of size $\ell \times \ell$. So $W_\ell$ is a $\ell^2 \times d$ matrix for some $d$ (usually smaller than $l^2$). Then, rotation of an $\ell \times \ell$ image $x$ by an angle $\theta$ is the $\ell^2$-size vector $W_\ell D_\ell(\theta) W_\ell^T x$, where $D_\ell$ is the block diagonal matrix described in section 2, of size $d \times d$. If a column of $W_\ell$ represents a basis vector which is invariant, the corresponding block is 1, and if a pair of columns in $W_\ell$ is a basis for an irreducible representation of type $n$, the block in $D_\ell$ for this pair is the elementary rotation matrix described in section 2. If there is no confusion we ignore the subscripts. We give three neural net architectures to discover CW-bases of images.

### 3.1 LEARNING CW-BASES

1. **An autoencoder(AE) architecture.** The autoencoder(AE) architecture is given as Alogrithm 1. An equivalent block diagram is given in the Appendix (see figure 6). Details of computing $Z_i$ below are given in Appendix, Section E.

---
**Algorithm 1:** AutoEncoder

---
**Input:** $X_i \in \mathbb{R}^{784}$, $i \in \{1, 2, \dots\}$
**Hyper-parameters :** $a_j, b_j, \beta$
Let $d_a = a_0 + \sum_{j=1} 2 \times a_j$ ; $\qquad d_b = b_0 + \sum_{j=1} 2 \times b_j$
**Output:** $W_{28} \in \mathbb{R}^{784 \times d_a}$
**Output:** $\phi \in \mathbb{R}^{d_b \times d_a}$          // Equivariant linear maps (Block diagonal
**Output:** $\psi \in \mathbb{R}^{d_a \times (d_b + d_b^2)}$          // matrices) as dictated by Remark 8

  **for** $i = 1, 2, \dots$ **do**
    |  $\theta_i \in U[0, 360)$               // Choosen uniformly at random
    |  $Y_i = \text{Rotate}(X_i, \theta_i)$       // Rotate is the standard image rotation
    |  $\hat{Y}_i = W_{28}^T Y_i$
    |  $Z_i = \psi(\phi(\hat{Y}_i) \otimes \phi(\hat{Y}_i) \oplus \phi(\hat{Y}_i))$
    |  $\hat{X}_i = W_{28} D(-\theta_i) Z_i$          // Unrotate and reconstruct the image
  **end**
  $L = \sum_i |X_i - \hat{X}_i|_2^2$                   // Loss function
  $W_{28}, \phi, \psi = \min_{W_{28}, \phi, \psi} (L + \beta |W_{28}^T W_{28} - Id_{d_a \times d_a}|_1)$
  **return** $W_{28}, \phi, \psi$

---

2. **Coupled Autoencoder(CAE) architecture**. In this setup, Figure 1, we learn $W_{14}$ and $W_{28}$ in tandem. We feed both, an image $X$ and a scaled down version of the image $x$ to the network. The network on top takes $x$ and produces $\hat{X}$ a 28x28 image. The bottom network takes $X$ and produces $\hat{x}$ a 14x14 image. The two networks are connected and we use the same $W_{28}$, and $W_{14}$ in the top and bottom layer. Hyperparameters are multiplicities of $SO(2)$-irreducible representations in $W_{28}$ and $W_{14}$. We learn the $W_{14}$, $W_{28}$ and two $SO(2)$-equivariant maps $\phi$ and $\psi$. The bottom network takes $X$, rotates it by $\theta$ and projects the resulting $Y$ on the current

$W_{28}$, to get $\hat{Y} = W_{28}^T Y$. Applying $\phi$ we get $\phi(\hat{Y})$. We unrotate $\phi(\hat{Y})$ using $D_{14}(-\theta)$ to get $z$. We generate a 14x14 image using the current $W_{14}$ to get $\hat{x} = W_{14}z$. The top network takes $x$ and rotates it by the same $\theta$. This is projected onto the current $W_{14}$ to get $\hat{y}$. We apply $\psi$ to $\hat{y} \oplus (\hat{y} \otimes \hat{y})$ and unrotate the resulting vector using $D_{28}(-\theta)$ to get $Z$. We use the current $W_{28}$ to get $\hat{X} = W_{28}Z$. We minimize the reconstruction error $|X - \hat{X}|^2 + |x - \hat{x}|^2$.

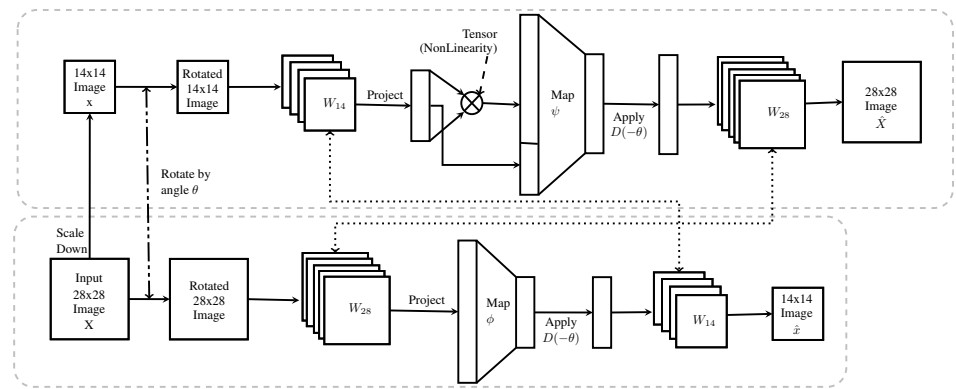

Figure 1: Coupled autoencoder architecture (CAE)

3 **Bootstrapping** $W_{14}$ **to** $W_{28}$, $W_{28}$ **to** $W_{14}$. The appendix describes an architecture in which we bootstrap, starting with a $W_{14}$ learned in experiment 1 or 2 and learn a $W_{28}$.

## 3.2 CLASSIFICATION USING THE LEARNED CW-BASIS

Figure 2 shows a 2 layer classification network for $28 \times 28$ images - this can be extended to any number of layers, and other image sizes. This is a logical extension of the auto encoder network. It is a CNN in Fourier domain using tensor product nonlinearity as in Thomas et al. (2018). Assume the dimension of $W_{28}$ is $d_W$. We project the image $x$ onto $W_{28}$ to get $W_{28}^T x$. Let $L_1$ be a vector space with an $SO(2)$-action - hyperparameters are multiplicities $l_{1j}$ of irreducibles $SO(2)$-representations $S_j$ in $L_1$. We learn an $SO(2)$-equivariant map $\phi_1$ from the image space of $W_{28}^T$ to $L_1$. The $SO(2)$-decomposition of $l_1 = \phi_1(W_{28}^T x) \oplus (\phi_1(W_{28}^T x) \otimes \phi_1(W_{28}^T x))$ is computed. This is the input to the next layer. There we choose a vector space $L_2$ with a $SO(2)$-action which splits into irreducibles $S_j$ with multiplicities $l_{2j}$ (hyperparameters). We learn a second $SO(2)$-equivariant map $\phi_2$ from $L_1 \oplus (L_1 \otimes L_1)$ to $L_2$ and compute $\phi_2(l_1)$. We compute the tensor product of $\phi_2(l_1)$ with itself and produce $l_2 = \phi_2(l_1) \oplus (\phi_2(l_1) \otimes \phi_2(l_1))$. This is the input to the next layer. So we learn as many $SO(2)$-equivariant maps as there are layers. In the final layer, $l_f$ is projected onto the invariant space (the multiplicity of $S_0$ in $L_f \oplus (L_f \otimes L_f)$ and fed to a soft max classifier.

## 3.3 RESULTS

1. **Construction of CW-basis**

   We construct CW-basis $W_{14}$ for $14 \times 14$ images and CW-basis $W_{28}$ for $28 \times 28$ in the AE and CAE architectures. We evaluate these bases in terms of image reconstruction error (MSE). We also used rotation reconstruction error by comparing with scikits-image

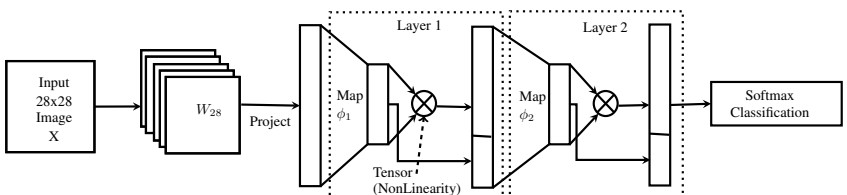

Figure 2: Classification network

rotation. The data set for which these errors are reported is MNIST-rot. We choose $[10, 5, 5, 5, 5, 4, 4, 4, 4, 4, 3, 3, 3, 3, 3, 2, 2, 2, 2, 2, 1, 1, 1, 1, 1]$ as our multiplicities for the $a_i$'s (hyperparameters) and $[8, 4, 4, 4, 4, 3, 3, 3, 3, 3, 2, 2, 2, 2, 2, 1, 1, 1, 1, 1]$ for the $b_i$'s (hyperparameters). Figure 3 and Figure 4 shows graphs of the error functions for $W_{28}$ constructed in AE, CAE architectures as a function of the number of samples given to learn the $W_{28}$. Even when the number of samples is as small as 50, we discover CW-bases which are good at rotation and reconstruction in both architectures. In Figure 7, in the Appendix we visualize a few basis elements learned.

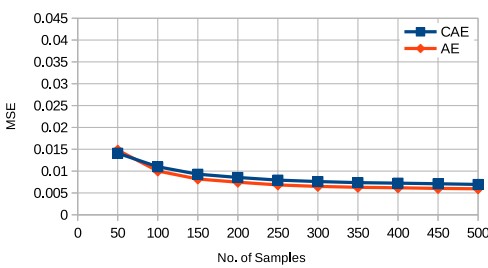 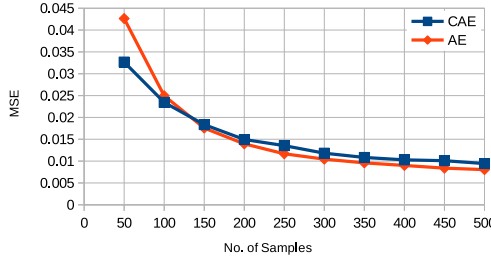

Figure 3: MSE - Reconstruction (MNIST-rot)     Figure 4: MSE - Rotation of images (MNIST-rot)

2. **Results on Classification, MNIST** In Figure 5 we plot the accuracy of these various $W_{28}$ obtained above when deployed for classification. We plot the accuracy of the classifier as a function of the number of samples used to construct the CW-bases. When the number of samples is as low as 50 a CAE-$W_{28}$ performs better than an AE-$W_{28}$. Beyond 100 samples the difference is insignificant. Plots using a bootstrapping CW-basis are similar to that of the AE and omitted.

In Table 1 we give the mean accuracy(and stdev) of our classifier when trained and tested on combinations using MNIST(NR) and MNIST-rot(R). The R/NR column for example denotes the accuracy when trained on MNIST-rot and tested on MNIST (LeCun et al.). For comparison we use Planar, Spherical CNN accuracies given in Cohen et al. (2018). We also compare with FFS2CNN, (see Kondor et al. (2018c)).

The fourth row shows the accuracy of our classifier that used a CAE-$W_{28}$. This was obtained using CAE architecture which was given 12000 training samples of MNIST-rot to discover the $W_{28}$. Our classifier performs better in all the scenarios.

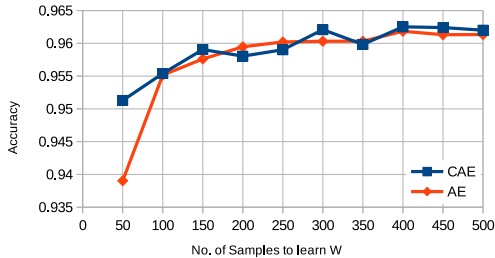

Figure 5: Classification accuracy (MNIST-rot)

**Coupling interchangeability** To test how coupled the CAE-$W_{28}$ and CAE-$W_{14}$ are, we did the following experiment - the classifier trained in row 4 with CAE-$W_{28}$ was presented with down sized 14x14 images for classification. No additional training was done. Instead we use the top half of the coupling network from Figure 1 - given a test image $y$ we compute $\hat{y} = W_{14}^T y$ using the CAE-$W_{14}$. Then $\psi((\hat{y} \otimes \hat{y}) \oplus \hat{y})$ is fed to the trained classifier of row 4. These results are reported in row 5 as [28/14 Tensor].

For comparison we took the 14x14 images and scaled them up to 28x28 and then fed them to the trained classifier of row 4. These results are reported in Row 6 as [28/14 Scale]. Using CAE-$W_{14}$ is better, indicating that a coupled bases retains scale information well.

We repeated the same experiment when the coupled network is given only 500 samples to learn the CAE-$W_{28}$ and CAE-$W_{14}$. As row 8 shows, performance is still very good.

Table 1: MNIST - Accuracies - rotated, unrotated combinations

|  | Samples used to learn W | R / R | R / NR | NR / NR | NR / R |
|---|---|---|---|---|---|
| Planar [1] | - | 23.00 | - | 98.00 | 11.00 |
| Spherical CNN [1] | - | 95.00 | - | 96.00 | 94.00 |
| FFS2CNN [2] | - | 95.80 | - | 96.00 | 95.86 |
| Ours | 12000 | 96.73 (0.35) | 96.90 (0.23) | 97.68 (0.08) | 98.48 (0.06) |
| 28 / 14 Tensor | 12000 | 96.17 (0.35) | 96.51 (0.27) | 97.10 (0.70) | 97.51 (0.09) |
| 28 / 14 Scale | 12000 | 94.79 (0.59) | 95.56 (0.55) | 93.86 (0.87) | 91.66 (1.36) |
| Ours | 500 | 96.40 (0.09) | 96.64 (0.06) | 97.41 (0.09) | 98.24 (0.05) |
| 28 / 14 Tensor | 500 | 95.78 (0.12) | 95.98 (0.09) | 96.53 (0.12) | 97.03 (0.07) |
| 28 / 14 Scale | 500 | 94.37 (0.39) | 95.34 (0.23) | 92.67 (1.02) | 89.62 (1.51) |

[1] Values as reported in Table 1 of Cohen et al. (2018)
[2] Values as reported in Kondor et al. (2018c)

3. **Results on Classification, Fashion-MNIST**. In Table 2 we report mean classification accuracy(and stdev) on the Fashion-MNIST data set(Xiao et al. (2017)). We rotate each data point around the origin by an angle chosen uniformly between 0 and $2\pi$ to create F-MNIST-rot. The CW basis was learned in the AE architecture with Fashion-MNIST as input. The classifier is a four layer network (96K parameters). We compared with a Depth 5 CNN (102K parameters).

Table 2: Fashion MNIST - Accuracies - rotated, unrotated combinations

|  | Samples used to learn W | R / R | R / NR | NR / NR | NR / R |
|---|---|---|---|---|---|
| CNN | - | 80.86 (0.57) | 79.83 (0.66) | 90.68 (0.31) | 20.86 (0.46) |
| Ours | 20000 | 86.34 (0.18) | 84.67 (0.27) | 86.70 (0.29) | 85.42 (0.18) |

## 3.4 IMPLEMENTATION

Our autoencoders and classifiers were implemented as neural nets in Python (TensorFlow). We used the Adam optimizer.

## 4 CONCLUSION AND FUTURE WORK

We answer in the affirmative all our motivating questions - we design architectures which learn CW-bases and classify using only tensor product nonlinearity. We obtain some state of the art results. It appears that coupling bases do retain scale information and can be interchangeably used for classification albeit with a small drop in performance. There is no appreciable advantage in using coupled bases for classification excepting when the bases are learned using small samples of inputs.

Tensor product nonlinearity allows us to decouple the classification problem in two steps. One advantage of decoupling the process is that the basis can be obtained offline. Our ideas can be extended to other groups. We have begun experiments with the action of the $S_n$ on images.

We are testing our architectures on CIFAR-10. One issue we run into when using architectures with a lot of depth, is that accuracy increases and then drops. A similar issue is resolved in RES-nets He et al. (2016) by computing residual functions. In ongoing work we proceed similarly since the residual function is $SO(2)$-equivariant.

Robustness of classifiers to noise is an active area of research, (see Goodfellow et al. (2018)). We believe our classifiers built from bases learnt in a CAE architecture should be robust to noise - our belief comes from the fact that coupling bases can be used interchangeably. We plan to investigate this.

### ACKNOWLEDGEMENTS

The authors would like to thank Infosys foundation for a generous grant.

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

# Appendices

## A  CONVOLUTION IN THE FOURIER DOMAIN

A CNN is equivariant to translations - i.e, whether we translate the image and convolve it with a filter or we convolve the image first and then translate it we get the same result. Thus the filters are naturally equivariant to the group of translations. We would like obtain such an equivariance with respect to the group of rotations.

To do so it is easier to work in the *Fourier space* of the group $SO(2)$. However a vector space basis in the Fourier space is not readily available - (this is what we call CW basis - Cohen and Welling compute this in their paper). Now every CW basis vector comes indexed by a non-negative integer (Paragraph 5 on Page 3). There could be multiple basis vectors indexed with the same integer which is called the multiplicity (Paragraph 6 of page 3). So in the Fourier space an image is a linear combination of CW basis vectors with coefficients (which of course depend upon the image). Lets call them CW coefficients. Denoting the CW basis by $W$, $\hat{x} = W^T x$ are the CW-coefficents of $x$. We first compute the CW basis given a reasonable number of samples of images $x$ and their rotations. The CW basis does not necessarily span the entire Fourier space, we need to find enough basis vectors which give a good approximation (Paragraph 7, page 3).

Convolution of an image with a $SO(2)$-equivariant filter means that whether we rotate an image and then convolve, or first convolve and then rotate we should end up transforming the image in the same way. This translates to the following in the Fourier domain - taking linear combinations of CW coefficients of CW-basis vectors of the same type m, thereby getting an equivariant linear map. The entries of this linear map $L$ is what we seek to find. So if we have a space V with a CW basis having $m_i$ basis vectors indexed by integer $i$ and another space W with $n_i$ basis vectors indexed by integers $i$, the search space for $SO(2)$-equivariant filters in the Fourier domain has dimension $\sum_i m_i n_i$, corresponding to block diagonal matrices, the $i$-th block being of size $n_i m_i$ (see paragraph 4 on page 3).

The natural nonlinearity in the Fourier space is multiplication of CW coefficients. When we multiply the coefficient of a basis vector of type m and the coefficient of a basis vector of type n, we get two coefficients, for basis vectors of type m+n and m-n (content of Remark 4). These are quadratic functions, of the starting CW coefficients - since this nonlinearity is obtained by taking tensor products of irreducible representations, we call it tensor product nonlinearity.

Our learning and classification takes place in the Fourier world.

## B  PROOFS

Here we prove that the bases obtained in the coupling architecture satisfy the definition of coupling. The proof is straightforward assuming that we have zero loss.

Let $W_{28}$ and $W_{14}$ be the final CW-bases learned. Following the bottom half of the network describing the coupling architecture, we see that under the morphism $\phi$, the image space of $W_{28}^T$ surjects onto the image space of $W_{14}^T$. Surjection follows since the dotted arrow going from $W_{14}$ on the top to $W_{14}$ below shows that we can start with any linear combination of the basis elements of $W_{14}$ at the top and obtain it as the image of $\phi$ applied to an appropriate linear combination of basis elements of $W_{28}$. So the image space of $W_{14}^T$ is isomorphic to a quotient of the image space of $W_{28}^T$. On the other hand following the top half of the network it is clear that $\psi$ maps the image space of $W_{14}^T \oplus (W_{14}^T \otimes W_{14}^T)$ surjectively onto the image space of $W_{28}^T$. Surjectivity follows by reasoning as above. Hence the bases obtained in the coupling architecture satisfy definition 6.

## C  LEARNING CW-BASIS

1. AutoEncoder(AE) architecture We describe this using figure Figure 6 (this architecture works for 14x14 images, 56x56 images). The hyperparameters are multiplicities with which $SO(2)$ irreducible representations occur in the images viewed as vectors in a 784 dimensional vector space with an action of $SO(2)$. We choose nonzero multiplicities $a_j$ for representations $S_j$ with

$0 \leq j \leq 24$ giving us a subspace of the image space of dimension $d_W = a_0 + \sum_j 2 * a_j$. We denote this subspace by $W_{28}$ and learn a CW-basis also denoted $W_{28}$, a $784 \times d_W$ matrix. We initialize $W_{28}$ randomly. We rotate a 28 x 28 image $x$ by a random angle $\theta$, to get $y$ and project $y$ onto $W_{28}$ to get $\hat{y} = W_{28}^T y$. We learn an $SO(2)$-equivariant map $\phi$ from the image space of $W_{28}^T$ to a vector space $L$ with an $SO(2)$-action, with multiplicities (hyperparameters) $b_j$. Consider $\phi(\hat{y}) \in L$. Now $L \otimes L$ gets a natural $SO(2)$ action. Recall that the tensor product of an $SO(2)$ irreducible representation of type $m$ and an $SO(2)$-irreducible representation of type type $n$ yields irreducible representations of types $m + n$ and $m - n$, (w.l.o.g $m > n$). We use this to compute the decomposition of $\phi(\hat{y}) \otimes \phi(\hat{y})$. The last set of parameters to be learned are an $SO(2)$-equivariant map $\psi$ from $L \oplus (L \otimes L)$ to the image space of $W_{28}^T$. Let $z = \psi(\phi(\hat{y}) \oplus (\phi(\hat{y}) \otimes \phi(\hat{y})))$. We unrotate $z$ using $D(-\theta)$. Then we reconstruct the image $\hat{x} = W_{28} D(-\theta) z$ We expect this to be close to $x$ so we minimize the reconstruction error $|\hat{x} - x|^2$ with a regularizer to ensure that $W_{28}^T W_{28}$ is identity.

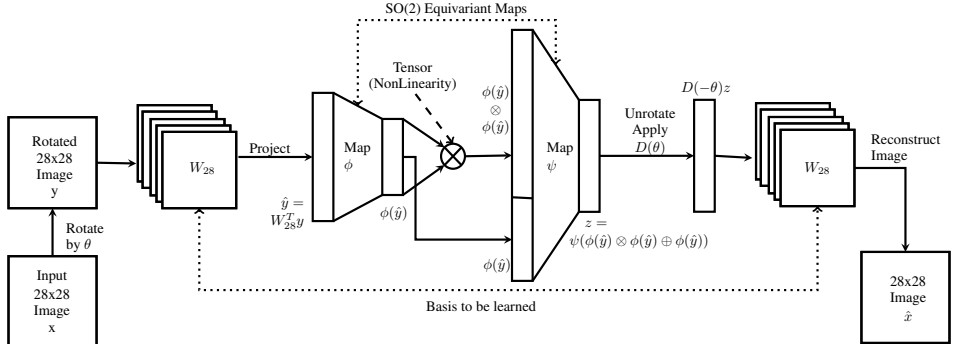

Figure 6: Autoencoder architecture (AE).

2. Bootstrapping architecture

In this experiment we start with a known $W_{14}$ obtained using experiment 1 for 14x14 images, to learn a $W_{28}$. The setup to go from $W_{14}$ to $W_{28}$ is exactly the part of the network in Figure 1 going from $X$ to $\hat{X}$ along the top. The hyperparameters are the multiplicities of $SO(2)$-irreducible representations in $W_{28}$. We learn the $SO(2)$-equivariant $\psi$ and $W_{28}$ by minimizing $|X - \hat{X}|^2$. Similarly starting with a known $W_{28}$ we can learn $W_{14}$ by following the path in Figure 1 from $x$ to $\hat{x}$ along the bottom.

## D  VISUALIZATION OF CW BASIS

Figure 7 shows a few basis elements learned using AE architucture on MNIST-rot.

## E  EXAMPLE

Let us see an example of how $Z_i$ is calculated in Algorithm 1. Let $d_a$ and $d_b$ be as given in the Algorithm. $\phi \in \mathbb{R}^{d_b \times d_a}$ is a block diagonal matrix with blocks of sizes $b_0 \times a_0, 2b_1 \times 2a_1, 2b_2 \times 2a_2, \ldots$ The first block has $(b_0 * a_0)$ entries ,the second block has $(b_1 * a_1)$ entries with the following structure.

$$\begin{pmatrix} \alpha_{11} & 0 & \alpha_{12} & 0 & \ldots & \alpha_{a_1 1} & 0 \\ 0 & \alpha_{11} & 0 & \alpha_{12} & \ldots & 0 & \alpha_{a_1 1} \\ \alpha_{21} & 0 & \alpha_{22} & 0 & \ldots & \alpha_{a_1 2} & 0 \\ 0 & \alpha_{21} & 0 & \alpha_{22} & \ldots & 0 & \alpha_{a_1 2} \\ \vdots & \vdots & \vdots & \vdots & \ddots & \vdots & \vdots \\ \alpha_{1 b_1} & 0 & \alpha_{2 b_1} & 0 & \ldots & \alpha_{a_1 b_1} & 0 \\ 0 & \alpha_{1 b_1} & 0 & \alpha_{2 b_1} & \ldots & 0 & \alpha_{a_1 b_1} \end{pmatrix}$$

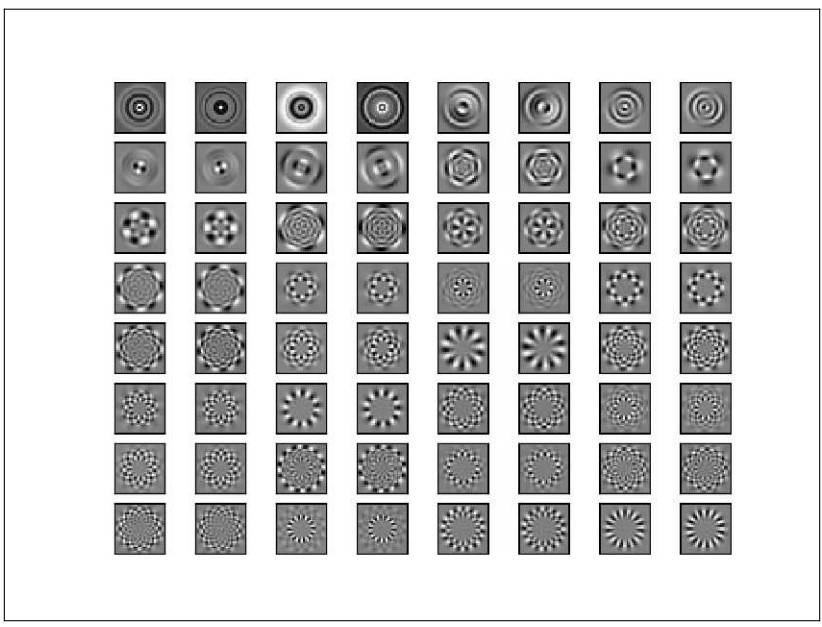

Figure 7: Basis for 28 x 28 images learned from MNIST-rot using AE

The multiplicities in $\hat{Y}$ are $a = [2, 1, 2]$ and that of $\phi(\hat{Y})$ are $b = [0, 1, 1]$. Hence $d_a = 8, d_b = 4$ and $\phi$ with arbitrary chosen values is given below.

$$\phi = \begin{pmatrix} 0 & 0 & \boxed{1 & 0} & 0 & 0 & 0 & 0 \\ 0 & 0 & \boxed{0 & 1} & 0 & 0 & 0 & 0 \\ 0 & 0 & 0 & 0 & \boxed{2 & 0 & 3 & 0} \\ 0 & 0 & 0 & 0 & \boxed{0 & 2 & 0 & 3} \end{pmatrix}$$

Boxes indicate the block for type 1 and type 2.

Let $\phi(\hat{Y}) = (p, q, r, s)^T$

When we tensor $(p, q)^T$ of type n with $(r, s)^T$ of type m (m > n), we get $(pr - qs, ps + qr)^T$ of type (m+n) and $(pr + qs, ps - qr)$ of type (m-n) (see Remark 4). In this example, when we tensor $\phi(\hat{Y})$ with itself, we get

$$\phi(\hat{Y}) \otimes \phi(\hat{Y}) = \begin{pmatrix} p^2 + q^2 \\ 0 \\ r^2 + s^2 \\ 0 \\ pr + qs \\ ps - qr \\ pr + qs \\ ps - qr \\ p^2 - q^2 \\ 2pq \\ pr - qs \\ ps + qr \\ pr - qs \\ ps + qr \\ r^2 - s^2 \\ 2rs \end{pmatrix}$$

with the following types : multiplicities - type 0 : 4, type 1 : 2, type 2 : 1, type 3 : 2, type 4 : 1.

During implementation we avoid duplicates and zeros and we use the following:

$$\phi(\hat{Y}) \otimes \phi(\hat{Y}) = \begin{pmatrix} p^2 + q^2 \\ r^2 + s^2 \\ pr + qs \\ ps - qr \\ p^2 - q^2 \\ 2pq \\ pr - qs \\ ps + qr \\ r^2 - s^2 \\ 2rs \end{pmatrix}$$

Also, $\phi(\hat{Y}) \otimes \phi(\hat{Y}) \oplus \phi(\hat{Y})$ after grouping the types, is given below:

$$\phi(\hat{Y}) \otimes \phi(\hat{Y}) \oplus \phi(\hat{Y}) = \begin{pmatrix} p^2 + q^2 \\ r^2 + s^2 \\ pr + qs \\ ps - qr \\ p \\ q \\ p^2 - q^2 \\ 2pq \\ r \\ s \\ pr - qs \\ ps + qr \\ r^2 - s^2 \\ 2rs \end{pmatrix}$$

The linear map $\psi \in \mathbb{R}^{d_c \times d_a}$ is similar to $\phi$. Only the input and output multiplicities are different. In the example we have $c = [2, 2, 2, 1, 1]$ and $a = [2, 1, 2]$. A particular $\psi$ with some fixed values is given below,

$$\psi = \begin{pmatrix} 1 & 2 & 0 & 0 & 0 & 0 & 0 & 0 & 0 & 0 & 0 & 0 & 0 & 0 \\ 3 & 4 & 0 & 0 & 0 & 0 & 0 & 0 & 0 & 0 & 0 & 0 & 0 & 0 \\ 0 & 0 & 5 & 0 & 6 & 0 & 0 & 0 & 0 & 0 & 0 & 0 & 0 & 0 \\ 0 & 0 & 0 & 5 & 0 & 6 & 0 & 0 & 0 & 0 & 0 & 0 & 0 & 0 \\ 0 & 0 & 0 & 0 & 0 & 0 & 7 & 0 & 8 & 0 & 0 & 0 & 0 & 0 \\ 0 & 0 & 0 & 0 & 0 & 0 & 0 & 7 & 0 & 8 & 0 & 0 & 0 & 0 \\ 0 & 0 & 0 & 0 & 0 & 0 & 9 & 0 & 10 & 0 & 0 & 0 & 0 & 0 \\ 0 & 0 & 0 & 0 & 0 & 0 & 0 & 9 & 0 & 10 & 0 & 0 & 0 & 0 \end{pmatrix}$$

And $Z_i = \psi(\phi(\hat{Y}) \otimes \phi(\hat{Y}) \oplus \phi(\hat{Y}))$

