# OpenReview forum: "Cohen Welling bases & SO(2)-Equivariant classifiers using Tensor nonlinearity."
_ICLR.cc/2019/Conference_

### Official Review · AnonReviewer2 · 2018-11-02
**An Important Problem, but insufficient experiments and unsure about some details of method**

**Rating:** 6
**Confidence:** 2

**Review:**

Review: This paper deals with the issue of learning rotation invariant autoencoders and classifiers.  While this problem is well motivated, I found that this paper was fairly weak experimentally, and I also found it difficult to determine what the exact algorithm was.  For example, how the optimization was done is not discussed at all.  At the same time, I'm not an expert in group theory, so it's possible that the paper has technical novelty or significance which I did not appreciate.

Strengths:

 -The challenge of learning rotation equivariant representations is well motivated and the idea of learning representations which transfer between different scales also seems useful.

Weaknesses:

-I had a difficult time understanding how the preliminaries (section 2) were related to the experiments (section 3).

-The reference (Kondor 2018) is used a lot but could refer to three different papers that are in the references.

  -Only reported results are on rotated mnist, but the improvements seem reasonable, but unless I'm missing something are worse than the 1.62% error reported by harmonic nets (mentioned in the introduction of the paper).  In addition to rot-mnist, harmonic nets evaluated boundary detection on the berkeley segmentation dataset.

  -It's interesting that the model learns to be somewhat invariant across scales, but I think that the baselines for this could be better.  For example, using a convolution network with mean pooling at the end, one could estimate how well the normal classifier handles evaluation at a different scale from that used during training (I imagine the invariance would be somewhat bad but it's important to confirm).


Questions:

-Section 3.1 makes reference to "learning parameters".  I assume that this is done in the usual way with backpropagation and then SGD/Adam or something?

-How is it guaranteed that W is orthogonal in the learning procedure?

---

> ### Author Response · Authors · 2018-11-11
> **Responses to AnonReviewer2**
>
> Thanks for the review comments.
> >>>Review: This paper deals with the issue of learning rotation invariant autoencoders and classifiers.  While this problem is well motivated, I found that this paper was fairly weak experimentally, and I also found it difficult to determine what the exact algorithm was.  For example, how the optimization was done is not discussed at all.  At the same time, I'm not an expert in group theory, so it's possible that the paper has technical novelty or significance which I did not appreciate.
>
> [Reply] The algorithm for learning a CW basis is now stated explicitly in the appendix. We have summarised our approach in the response to AnonReviewer3. We have given it as comments titled "Summary 1 of 2" & "Summary 2 of 2".
>
> We discuss our Implementation now in Section 3.4. The main technical novelty is that equivariance is easily learned in the CW basis. As AnonReviewer1 points out, tensor product nonlinearity is perhaps more important than the basis itself.
> -----
>
> >>>Strengths:
>
> >>> -The challenge of learning rotation equivariant representations is well motivated and the idea of learning representations which transfer between different scales also seems useful.
>
> [Reply] Thanks for this encouragement.
> -----
>
> >>>Weaknesses:
>
> >>>-I had a difficult time understanding how the preliminaries (section 2) were related to the experiments (section 3).
>
> [Reply] Sorry about this. Perhaps a reason for confusion is that whereas we use the phrase G-morphism
> in Section 2, we use the phrase SO(2) equivariant maps in Section 3. These are the same.
> -----
>
> >>>-The reference (Kondor 2018) is used a lot but could refer to three different papers that are in the references.
>
> [Reply] Sorry about this. This is now corrected.
> -----
>
> >>>  -Only reported results are on rotated mnist, but the improvements seem reasonable, but unless I'm missing something are worse than the 1.62% error reported by harmonic nets (mentioned in the introduction of the paper).  In addition to rot-mnist, harmonic nets evaluated boundary detection on the berkeley segmentation dataset.
>
> [Reply] Yes we get about 97%, less than what harmonic nets get but the architecture is very simple. One aspect we did not emphasise much is the last column in table 1. It is known that Harmonic nets (and many other equivariant networks) need a large amount of data augmentation to perform well on MNIST-rot when trained on upright MNIST. We need no such augmentation once we have a reasonable W_{28}. In that sense our network is like spherical and FFS2CNN - truly rotation equivariant.
>
> We will take a look at Berkeley segmentation data and see what harmonic nets do and see if we can conduct those experiments. Thanks for this suggestion.
> -----
>
> >>>  -It's interesting that the model learns to be somewhat invariant across scales, but I think that the baselines for this could be better.  For example, using a convolution network with mean pooling at the end, one could estimate how well the normal classifier handles evaluation at a different scale from that used during training (I imagine the invariance would be somewhat bad but it's important to confirm).
>
> [Reply] Thanks for this suggestion. We have run these experiments. We trained a CNN with about 489K parameters on MNIST-rot 28x28 images, getting a 95.1 percent accuracy.
> When this was fed 14x14 images scaled up to 28x28, we get 90.5 percent accuracy. Should we report this in the main paper?
> -----
>
> >>>Questions:
>
> >>>-Section 3.1 makes reference to "learning parameters".  I assume that this is done in the usual way with backpropagation and then SGD/Adam or something?
>
> [Reply] Yes, backpropogation using ADAM optimiser. We make this explicit in Section 3.4
> -----
>
> >>>-How is it guaranteed that W is orthogonal in the learning procedure?
>
> [Reply] Sorry, we should have mentioned this -we do so now - we add a regularizer to the reconstruction loss.
> -----

---

> > ### Comment · AnonReviewer2 · 2018-11-18
> > **More details in algorithmic block**
> >
> > Thanks for responding.  Can you give more details on how Z_i is computed in the algorithmic block?

---

> > > ### Author Response · Authors · 2018-11-21
> > > **Details about computing Z_i**
> > >
> > > We have added the details in the appendix of the current revision of the paper. Please let us know if you need more information.

---

### Official Review · AnonReviewer1 · 2018-11-03
**A bit rough around the edges, but there are some interesting lessons to be learned here if one reads between the lines.**

**Rating:** 7
**Confidence:** 4

**Review:**

Recently there has been a spate of work on generalized CNNs that are equivariant to various symmetry groups, such a 2D and 3D rotations, the corresponding Euclidean groups (comprising not just rotations but also translations) and so on. The approach taken in most of the recent papers is to explicitly build in these equivariances by using the appropriate generalization of convolution. In the case of nontrivial groups this effectively means working in Fourier space, i.e., transforming to a basis that is adapted to the group action. This requires some considerations from represntation theory.

Earlier, however, there was some less recognized work by Cohen and Welling on actually learning the correct basis itself from data. The present paper takes this second approach, and shows for a simple task like rotated MNIST, the basis can be learned from a remarkably small amount of data, and actually performs even better than some of the fixed basis methods. There is one major caveat: the nonlinearity itself has to be rotation-covariant, and for this purpose they use the recently introduced tensor product nonlinearities.

The paper is a little rough around the edges. In the first 4 pages it launches into an exposition of ideas from representation theory which is too general for the purpose: SO(2) is a really simple commutative group, so the way that "tensor product" representations reduce to irreducibles could be summed up in the formula  $e^{-2\pi i k_1 x}e^{-2\pi i k_2 x}=e^{-2\pi i (k_1+k_2) x}$. I am not sure why the authors choose to use real representations (maybe because complex numbers are not supported in PyTorch, but this could easily be hacked) and I find that the real representations make things unnecessarily complicated. I suspect that at the end of the day the algorithm does something very simple (please clarify if working with
real representations is somehow crucial).

But this is exactly the beauty of the approach. The whole algorithm is very rough, there are only two layers (!), no effort to carefully implement nice exact group convolutions, and still the network is as good as the competition. Another significant point is that this network is only equivariant to rotations and not translations.

Naturally, the question arises why one would want to learn the group adapted basis, when one could just compute it explicitly. There are two interesting lessons here that the authors could emphasize more:

1. Having a covariant nonlinearity is strong enough of a condition to force the network to learn a group adapted (Cohen-Welling) basis. This is interesting because Fourier space ("tensor") nonlinearities are a relatively new idea in the literature. This finding suggests that the nonlinearity might actually be more important than the basis.

2. The images that the authors work on are not functions on R^2, but just on a 28x28 grid. Rotating a rasterized image with eg. scikit-rotate introduces various artifacts. Similarly, going back and forth between a rasterized and polar coordinate based representation (which is effectively what would be required for "Harmonic Networks" and other Fourier methods) introduces messy interpolation issues. Not to mention downsampling, which is actually addressed in the paper. If a network can figure out how to best handle these issues from data, that makes things easier.

The experiments are admittedly very small scale, although some of the other publications in this field also only have small experiments. At the very least it would be nice to have standard deviations on the results and some measure of statistical significance. It would be even nicer to have some visualization of the learned bases/filters, and a bare bones matrix-level very simple description of the algorith. Again, what is impressive here is that such a small network can learn to do this task reasonably well.

Suggestions:

1. Also cite the Tensor Field Networks of Thomas et al in the context of tensor product nonlinearities.

2. Clean up the formatting. "This leads us to the following" in a line by itself looks strange. Similarly "Classification ising the learned CW-basis". I think something went wrong with \itemize in Section 3.1.

---

> ### Author Response · Authors · 2018-11-11
> **Responses to AnonReviewer1**
>
> Thanks for the review comments.
> >>>The paper is a little rough around the edges. In the first 4 pages it launches into an exposition of ideas from representation theory which is too general for the purpose: SO(2) is a really simple commutative group, so the way that "tensor product" representations reduce to irreducibles could be summed up in the formula  $e^{-2\pi i k_1 x}e^{-2\pi i k_2 x}=e^{-2\pi i (k_1+k_2) x}$. I am not sure why the authors choose to use real representations (maybe because complex numbers are not supported in PyTorch, but this could easily be hacked) and I find that the real representations make things unnecessarily complicated. I suspect that at the end of the day the algorithm does something very simple (please clarify if working with real representations is somehow crucial).
>
> [Reply] Yes what you say is absolutely correct - that we needn't have presented it in this generality. But one reason to do this was to suggest that the same idea will work for other groups also if that group acts naturally on objects like we have SO(2) acting on images. All you need is to understand how tensor products of irreducibles split for that group. As we mention in the conclusion we have begun exploring with the symmetric group.
>
> We have implemented our algorithms in the complex world also and the results are almost the same. However since we are using tensor flow we decided to work with reals.  And since we wished to highlight the Cohen -Welling paper as one of our inspirations, we work over reals following what Cohen and Welling do.
> -----
>
> >>>But this is exactly the beauty of the approach. The whole algorithm is very rough, there are only two layers (!), no effort to carefully implement nice exact group convolutions, and still the network is as good as the competition. Another significant point is that this network is only equivariant to rotations and not translations.
>
> [Reply] Thanks for these encouraging comments.
> -----
>
> >>>1. Having a covariant nonlinearity is strong enough of a condition to force the network to learn a group adapted (Cohen-Welling) basis. This is interesting because Fourier space ("tensor") nonlinearities are a relatively new idea in the literature. This finding suggests that the nonlinearity might actually be more important than the basis.
>
> [Reply] Thanks for making this so explicit. We will include this in our paper. Please refer to the explanation given to AnonReviewer3, where we summarise our work and point to this remark of yours.
> -----
>
> >>>2. The images that the authors work on are not functions on R^2, but just on a 28x28 grid. Rotating a rasterized image with eg. scikit-rotate introduces various artifacts. Similarly, going back and forth between a rasterized and polar coordinate based representation (which is effectively what would be required for "Harmonic Networks" and other Fourier methods) introduces messy interpolation issues. Not to mention downsampling, which is actually addressed in the paper. If a network can figure out how to best handle these issues from data, that makes things easier.
>
> [Reply] Again, thanks for the encouraging comments. We will emphasize these points.
> -----
>
> >>>The experiments are admittedly very small scale, although some of the other publications in this field also only have small experiments. At the very least it would be nice to have standard deviations on the results and some measure of statistical significance. It would be even nicer to have some visualization of the learned bases/filters, and a bare bones matrix-level very simple description of the algorithm. Again, what is impressive here is that such a small network can learn to do this task reasonably well.
>
> [Reply] Thanks for this suggestion. We have given a separate table with some statistics of our experiments - our earlier table reported accuracies in the scale 0 to 1 but deviations are better expressed in percentage. So we have put a new table. Should we replace the earlier table with the new table (adding the percentage accuracies of the baseline models)? And pictures of filters are now in the appendix. And again thanks for appreciating that a small network suffices. We have a complete description of Experiment 1 as an algorithm in the appendix now. Should we put this in place of the current text?
>
> -----
>
> >>>Suggestions:
>
> >>>1. Also cite the Tensor Field Networks of Thomas et al in the context of tensor product nonlinearities.
>
> [Reply] Thanks for pointing this out. We will make an explicit reference to this in the next revision.
> ------
>
> >>>2. Clean up the formatting. "This leads us to the following" in a line by itself looks strange. Similarly "Classification ising the learned CW-basis". I think something went wrong with \itemize in Section 3.1.
>
> [Reply] Sure. Sorry for this. It has been cleaned up.
> -----

---

### Official Review · AnonReviewer3 · 2018-11-05
**Difficult to read, insufficient evaluation**

**Rating:** 3
**Confidence:** 2

**Review:**

This paper proposes autoencoder architectures based on Cohen-Welling bases for learning rotation-equivariant image representations. The models are evaluated by reconstruction error and classification in the space of the resulting basis on rotated-MNIST, showing performance improvements with small numbers of parameters and samples.

I found most of this submission difficult to read and digest. I did not understand much of the exposition. I’ll freely admit I haven’t followed this line of work closely, and have little background in group theory, but I doubt I’m much of an outlier among the ICLR audience in that regard. The “Preliminaries” section is very dense and provides little hand-holding for the reader in the form of context, intuition, or motivation for each definition and remark it enumerates. I can't tell how much of the section is connected to the proposed models. (For comparison, I skimmed the prior work that this submission primarily builds upon (Cohen & Welling, 2014) and found it relatively unintimidating. It gently introduces each concept in terms that most readers familiar with common machine learning conventions would be comfortable with. It's possible to follow the overall argument and get the "gist" of the paper without understanding every detail.)

All that being said, I don’t doubt this paper makes some interesting and important contributions -- I just don’t understand what they are.

Here are some specific comments and questions, mostly on the proposed approaches and experiments:

* What actually is the “tensor (product) nonlinearity”? Given that this is in the title and is repeatedly emphasized in the text, I expected that it would be presented much more prominently. But after reading the entire paper I’m still not 100% sure what “tensor nonlinearity” refers to.

* Experiments: all models are described in long-form prose. It’s very difficult to read and follow. This could be made much clearer with an algorithm box or similar.

* The motivation for the “Coupled Autoencoder” model isn’t clear. What, intuitively, is to be gained from reconstructing a high-resolution image from a low-resolution basis and vice versa? The empirical gains are marginal.

* Experiments: the structure of the section is hard to follow. (1) and (2) are descriptions of two different models to do the same thing (autoencoding); then (3) (bootstrapping) is another step done on top of (1), and finally (4) is a classifier, trained on top of (1) or (2). This could benefit from restructuring.

* There are long lists of integer multiplicities a_i and b_i: these seem to come out of nowhere, with no explanation of how or why they were chosen -- just that they result in “learn[ing] a really sharp W_28”. Why not learn them?

* How are the models optimized? (Which optimizer, hyperparameters, etc.?)

* The baseline methods should also be run on the smaller numbers of examples (500 or 12K) that the proposed approach is run on.

* A planar CNN baseline should be considered for the autoencoder experiments.

* Validating on MNIST alone (rotated, spherical, or otherwise) isn’t good enough in 2018. The conclusions section mentions testing the models with deeper nets on CIFAR, but the results are not reported -- only hinting that it doesn’t work well. This doesn’t inspire much confidence.

* Why are Spherical CNNs (Cohen et al., 2018) a good baseline for this dataset? The MNIST-rot data is not spherical.

* Table 1: The method labels (Ours, 28/14 Tensor, and 28/14 Scale) are not very clear (though they are described in the text)

* Table 1: Why not include the classification results for the standard AE? (They are in the Fig. 6 plot, but not the table.)

* Conclusions: “We believe our classifiers built from bases learnt in a CAE architecture should be robust to noise” -- Why? No reasons are given for this belief.

* There are many typos and grammatical errors and odd/inconsistent formatting (e.g., underlined subsection headers) throughout the paper that should be revised.

---

> ### Author Response · Authors · 2018-11-11
> **Responses to AnonReviewer3 - 1 of 2**
>
> Thanks for the review comments.
> >>>I found most of this submission difficult to read and digest. I did not understand much of the exposition. .........I don’t doubt this paper makes some interesting and important contributions -- I just don’t understand what they are.
>
> [Reply] We are sorry that the you found the paper difficult to read. We think one source for confusion could be that we never explicitly stated that the term G-morphism used in Section 2 is the same as equivariant map used in Section 3. We do so now.
>
> We have a short summary of what we do in comments titled "Summary 1 of 2" using a language more familiar to the ML community. Hope this helps. In "Summary 2 of 2" (again short:) we show how we apply this. We could try and incorporate this into the main paper.
> -----
>
> >>>Here are some specific comments and questions, mostly on the proposed approaches and experiments:
>
> >>>* What actually is the “tensor (product) nonlinearity”? Given that this is in the title and is repeatedly emphasized in the text, I expected that it would be presented much more prominently. But after reading the entire paper I’m still not 100% sure what “tensor nonlinearity” refers to.
>
> [Reply] We hope this is answered in the explanation given in comments titled "Summary 1 of 2".
> -----
>
> >>>* Experiments: all models are described in long-form prose. It’s very difficult to read and follow. This could be made much clearer with an algorithm box or similar.
>
> [Reply] Since we were referring to diagrams to explain the algorithm we felt it was easier to follow it this way.  However we have written Experiment 1 as an algorithm as suggested by you and AnonReviewer1. Currently it is in the appendix. Please  let us know if this should replace the long text.
> -----
>
> >>>* The motivation for the “Coupled Autoencoder” model isn’t clear. What, intuitively, is to be gained from reconstructing a high-resolution image from a low-resolution basis and vice versa? The empirical gains are marginal.
>
> [Reply] That the abstract elementary features learned from such a basis should be invariant to scale is the motivation for defining this. When we started we expected that features learned from this basis would be superior at classification, but our experiments show that is not the case. However the coupled bases could be used interchangeably as we show in  Section 3.3 Results of classification, Coupling interchangeability. Our experiments also show that we can simultaneously learn Fourier bases in different scales, which can later deal with downsampled images.
>
> As an application: - Consider the problem of farmers having to deal with pests which they do not recognize, but limited by resources of bandwidth, and not having cellphones which take high resolution images. A solution would be to have a high end server deployed at a central location which is trained to recognise pests using say the basis from a Coupled autoencoder. When a farmer sees a new pest she could take a photograph of this on her cell phone and transmit this low resolution image to the server - the server can then use our model. (This needs to be tested on real world examples, something we hope to take up)
> -----
>
> >>>* Experiments: the structure of the section is hard to follow. (1) and (2) are descriptions of two different models to do the same thing (autoencoding); then (3) (bootstrapping) is another step done on top of (1), and finally (4) is a classifier, trained on top of (1) or (2). This could benefit from restructuring.
>
> [Reply] Thanks for the suggestion - We have restructured it a little by giving subsection headings and we have rewritten some parts.
> -----
>
> >>>* There are long lists of integer multiplicities a_i and b_i: these seem to come out of nowhere, with no explanation of how or why they were chosen -- just that they result in “learn[ing] a really sharp W_28”. Why not learn them?
>
> [Reply] These are hyperparameters fine tuned by us - how many CW basis vectors to choose which are indexed by integers 0, 1,..., 24 respectively.  As for learning them, yes we could try learning them and would like to do carry out experiments to see if that works.
> -----
>
> >>>* How are the models optimized? (Which optimizer, hyperparameters, etc.?)
>
> [Reply] We mention this now explicitly in an Implementation Section 3.4
>
> We use Adam optimiser and implement all of this in Tensorflow. We just used what tensor flow offers with no modification. Everywhere hyperparameters are multiplicities of irreducible representations in the domain and range of our SO(2) equivariant maps \psi and \phi . We do mention hyperparameters explicitly in all our experiments.
> -----

---

> ### Author Response · Authors · 2018-11-11
> **Responses to AnonReviewer3 - 2 of 2**
>
>
> >>>* The baseline methods should also be run on the smaller numbers of examples (500 or 12K) that the proposed approach is run on.
>
> [Reply] Sorry, we are not clear about what you mean - since we have decoupled learning the bases and using it for classification, we are not sure what it will mean to run the the baselines on 500 samples - since when deploying for classification we train our classifier on the full training set of 12K samples of MNIST-ROT. What we are only saying is that a good bases for reconstruction can be learned with 500 input samples. And the projected linear CW coefficients they give us are good for classification - our classifier constructs (upto) degree six polynomials in these coefficients and then does a softmax classification.
> -----
>
> >>>* A planar CNN baseline should be considered for the autoencoder experiments.
>
> [Reply] Sorry, again it is not clear to us what you mean. A CNN auto encoder will probably settle down and reconstruct the image well. But we will need to train it for classification - it is not clear to us how to use the parameters that such a CNN autoencoder learns during reconstruction, when it is deployed for classification.
> -----
>
> >>>* Validating on MNIST alone (rotated, spherical, or otherwise) isn’t good enough in 2018. The conclusions section mentions testing the models with deeper nets on CIFAR, but the results are not reported -- only hinting that it doesn’t work well. This doesn’t inspire much confidence.
>
> [Reply] Yes, we agree that we need to do more.  However we started focussing on experiments with symmetric group representations mentioned in the conclusions because this hasn't been studied much, and has some interesting connections. That said we will surely take up your suggestion and resume working with CIFAR.
> -----
>
> >>>* Why are Spherical CNNs (Cohen et al., 2018) a good baseline for this dataset? The MNIST-rot data is not spherical.
>
> [Reply] The authors embed MNIST-rot images on a sphere and then test their model. Their models and ours do their learning in the Fourier space.
> -----
>
> >>>* Table 1: The method labels (Ours, 28/14 Tensor, and 28/14 Scale) are not very clear (though they are described in the text)
>
> [Reply] Thanks for the suggestion. We have expanded on this. Please let us know if this reads better.
> -----
>
> >>>*Table 1: Why not include the classification results for the standard AE? (They are in the Fig. 6 plot, but not the table.)
>
> [Reply] Since the accuracies of our AE are similar to that of our CAE we have not included it. We could certainly do it. Please let us know your feedback on this.
> -----
>
> >>>* Conclusions: “We believe our classifiers built from bases learnt in a CAE architecture should be robust to noise” -- Why? No reasons are given for this belief.
>
> [Reply] We believe the coupled bases will be robust because they do work well on downsampled images -  of course all this needs to be tested. It is something we would like to do.
> -----
>
> >>>* There are many typos and grammatical errors and odd/inconsistent formatting (e.g., underlined subsection headers) throughout the paper that should be revised.
>
> [Reply] Sorry about this. We have revised it accordingly and we think all typos are now taken care off.
> -----

---

> ### Author Response · Authors · 2018-11-11
> **Summary 1 of 2**
>
>
> Summary of Work:
>
> A conventional neural network is equivariant to translations - i.e whether we translate the image and convolve it with a filter or we convolve the image first and then translate it we get the same result. We wish to implement this with rotations - to do this it is easier to work in the "Fourier space" of the group SO(2), which are functions on irreducible representations of SO(2). However a vector space basis in the Fourier space is not readily available - (this is what we call CW basis - Cohen and Welling compute one in their paper). Now every CW basis vector comes indexed by a non-negative integer (Paragraph 5 on Page 3). There could be multiple basis vectors indexed with the same integer which is called the multiplicity (Paragraph 6 of page 3). So in the Fourier space an image is a linear combination of CW basis vectors with coefficients (which depend upon the image). Lets call them CW coefficients. We don't need the CW basis to span the entire Fourier space, it suffices to find enough basis vectors which give a good approximation (Paragraph 7, page 3).
>
> This is what we first compute, given a reasonable number of samples of the images and their rotations.
>
> Convolving an image by a G-equivariant filter means that whether we rotate an image and then convolve or first convolve with the filter and then rotate has same effect. This  translates to the following in the Fourier domain - taking linear combinations of CW coefficients of CW-basis vectors of the same type m. The variables of this filter are entries of this linear map. So if we have a Fourier space V and a CW basis with $m_i$ basis vectors indexed by integer $i$ and another Fourier space W with $n_i$ basis vectors indexed by integers $i$, the search space for G-equivariant filters in the Fourier domain is of dimension $\sum_i m_i n_i$ - corresponding to block diagonal matrices, the i-th block being of size n_i m_i (paragraph 4 on page 3).
>
> The natural nonlinearity in the Fourier space is multiplication of CW coefficients. When we multiply the coefficient of a basis vector of type m and the coefficient of a basis vector of type n, we get two coefficients, for basis vectors of type m+n and m-n (content of Remark 4). These are quadratic functions, NOT linear functions of the starting CW coefficients - since this nonlinearity is obtained from tensor product of irreducible representations we call it tensor product nonlinearity.
>
> All our learning happens in the Fourier world.

---

> ### Author Response · Authors · 2018-11-11
> **Summary 2 of 2**
>
>
> Learning CW basis:
>
> In the experiment 1 to construct a CW basis our hyperparameters are [a_0,a_1,..,a_24] , a_i denoting the number of CW basis indexed by i in the Fourier space of input images.  Likewise b=[b_0,..] are hyperparameters of the Fourier space of the range V of the filter (equivariant map) $\phi$. So \phi is an appropriate sized block diagonal matrix taking us in an equivariant manner from the Fourier space of images to V. And we need to learn \phi.
>
> Having decided b_i ,we know from Remark 4 the multiplicities of CW basis indexed by i in the Fourier domain of the vector space V \otimes V. We come down from the Fourier space of  V \oplus (V \otimes V) to the Fourier space of the input images by a filter (equivariant map) \psi. So the composite map denoted $\psi(phi(\hat{y}) \ otimes phi(\hat{y}))  \oplus \phi(\hay{y}))$ is a filter from the Fourier space of  input images back to itself. The variables of the filter are the entries of the block diagonal matrices describing \psi and \phi. And this composite map is a nonlinear function of the CW coefficients \hat{y}, obtained using tensor product.
>
> We do not know the Fourier basis W_{28} of input images to begin with. Given about 500, 28 x 28 images we can discover a good Fourier basis which works for all images. Please refer to the algorithm given now in the appendix. Setting a_i and b_i as given gives us the Fourier basis W_{28}, whose plots are given in Figures 4,5,6.
>
> The second and third experiments are similar. But we are learning more filters (equivariant maps) in the second experiment since we are learning a W_{14} also.
>
> As AnonReviewer1 explicitly points out, we do not compute the CW basis vectors of the intermediate vector space V nor of V \otimes V. We only need to compute the CW basis of the image space and use tensor product nonlinearity.
>
>
> Classification:
>
> The set of CW coefficients of an image are its "elementary linear features" (paragraph 8 of page 3). We can construct more abstract features by multiplying these coefficients and taking linear combinations of coefficients indexed by the same integer n.
>
> Hyperparameters are L_1=[l_10,l_11,..] the multiplicities of CW basis of the Fourier space of the range of \phi_1 and L_2=[l_20,.], multiplicities of the range of \phi_2. And once again filters  (equivariant maps) $\phi_1, \phi_2, to be learned are block diagonal linear transformations.
>
> All of this is implemented as neural networks in tensor flow. We can release our code anytime.
>
> Implementation details are stated explicitly now in Section 3.4
>
> Again we only need a good vector space basis of the Fourier space of input images -  we don’t need  to explicitly compute a CW basis of the intermediate Fourier spaces we encounter. And again we use tensor products to compute more abstract features.

---

### Author Response · Authors · 2018-11-26
**Added experimental results on Fashion-MNIST**

We have experimented our algorithms on Fashion-MNIST dataset and reported the results in the current revision of the paper.

---

### Meta-Review · Area_Chair1 · 2018-12-16
**Interesting ideas, but currently not sufficiently well presented**

**Confidence:** 5
**Recommendation:** Reject

**Metareview:**

This paper studies group equivariant neural network representations by building on the work by [Cohen and Welling, '14], which introduced learning of group irreducible representations, and [Kondor'18], who introduced tensor product non-linearities operating directly in the group Fourier domain.

Reviewers highlighted the significance of the approach, but were also unanimously concerned by the lack of clarity of the current manuscript, making its widespread impact within ICLR difficult, and the lack of a large-scale experiment that corroborates the usefulness of the approach. They were also very positive about the improvements of the paper during the author response phase. The AC completely agrees with this assessment of the paper. Therefore, the paper cannot be accepted at this time, but the AC strongly encourages the authors to resubmit their work in the next conference cycle by addressing the above remarks (improve clarity of presentation and include a large-scale experiment).